# FairSAM: Fair Classification on Corrupted Image Data Through Sharpness-Aware Minimization

**Yucong Dai**\*  *yucongd@clemson.edu*
*Clemson University*

**Jie Ji**\*  *jji@clemson.edu*
*Clemson University*

**Xiaolong Ma**  *xiaolom@clemson.edu*
*Clemson University*

**Yongkai Wu**  *yongkaw@clemson.edu*
*Clemson University*

**Reviewed on OpenReview:** *https://openreview.net/forum?id=W2QKvn57yw*

## Abstract

Image classification models trained on clean data often degrade sharply when exposed to corrupted test or deployment data, such as images with impulse noise, Gaussian noise, or environmental noise. This degradation reduces overall performance and disproportionately affects demographic subgroups, raising algorithmic bias concerns. Although robust learning algorithms such as Sharpness-Aware Minimization improve overall robustness and generalization, they do not address biased performance degradation across demographic subgroups. Existing fairness-aware machine learning methods reduce performance disparities but struggle to maintain robust and equitable accuracy across demographic subgroups under data corruption. This limitation reveals an inherent tension between robustness and fairness under corrupted data. To address these challenges, we introduce a metric to assess performance degradation across subgroups under data corruption. We propose **FairSAM**, a framework that integrates Fairness-oriented strategies into SAM to equalize performance across demographic groups under corrupted conditions. Experiments on multiple real-world datasets and prediction tasks show that FairSAM balances robustness and fairness in corrupted image classification. The framework yields a structured solution for fair and robust image classification in the presence of data corruption.

## 1 Introduction

Deep neural networks have achieved strong results in AI applications such as image classification, image segmentation, and object detection. However, corrupted data challenge deep neural networks in real-world applications. In this paper, we focus on image classification tasks that encounter corrupted data from diverse sources, including user-uploaded images, mobile device captures, and network transmission artifacts. We investigate unequal model degradation when models trained on clean data are applied to corrupted images affected by impulse noise, Gaussian noise, weather effects (e.g., snow, fog), and motion blur during capture or transfer. Recent work (Nanda et al., 2021) shows that this degradation disproportionately affects demographic subgroups, i.e., accuracy drops vary significantly across subgroups and raise algorithmic bias concerns. Thus, applying models trained on clean data to corrupted images can create or amplify algorithmic bias.

---

\*Equal contribution

Although various fairness-aware methods, such as fairness constraints (Calders & Verwer, 2010; Kamiran et al., 2010; Corbett-Davies et al., 2017; Zafar et al., 2017b) and reweighing strategies (Calders et al., 2009), have been proposed to address bias in machine learning models, they often fail to mitigate unequal accuracy degradation across subgroups and neglect broader robustness requirements.

Recent research frames corrupted-data classification as a robustness and generalization challenge as the model must maintain performance under noise perturbations that differ from the training distribution. Sharpness-Aware Minimization (SAM) (Foret et al., 2021) has emerged as an effective robustness approach by promoting "flat" minima in the loss landscape, where loss changes gradually with parameter variation. This property improves generalization and resilience to data corruption. However, while SAM improves aggregate robustness, it does not inherently address fairness across demographic subgroups. SAM's performance gains are often unevenly distributed, and some disadvantaged subgroups remain more susceptible to accuracy degradation. This disparity highlights a critical limitation of SAM in scenarios where both robustness and fairness are equally essential.

To address these inherent challenges, we first formulate the fair classification problem in the context of corrupted data. **Specifically, we study a setting where a model is trained on clean data and tested on corrupted data containing various types of noise.** We restrict the current scope to binary sensitive attributes, where each sensitive attribute partitions samples into advantaged and disadvantaged groups. This setting reflects real deployments where training data are carefully curated but trained models face unexpected corrupted data in the wild. We then evaluate model performance and assess performance degradation across demographic subgroups, such as age (young/non-young) and gender (female/male), to understand both robustness and fairness under corrupted conditions. We introduce a metric to quantify fairness in performance degradation under corruption. This metric differs from one-shot fairness notions that require equal robustness across population partitions under imperceptible input perturbations. The metric, *Corrupted Degradation Disparity*, captures the difference in accuracy degradation (i.e., the drop in accuracy between clean and corrupted data) between specific subgroups, such as young and non-young individuals. Based on this metric, we propose **FairSAM**, the first framework that incorporates fairness-oriented strategies into SAM. Specifically, we develop instance-reweighted SAM and approximate per-sample perturbations with a per-batch perturbation algorithm to promote fairness and robustness simultaneously. FairSAM distributes robustness improvements equitably across demographic subgroups. It addresses fairness concerns while maintaining high overall accuracy under corrupted conditions.

We conduct experiments on multiple real-world datasets, including imbalanced CelebA and balanced Fair-Face, across prediction tasks with different target and sensitive attributes. These experiments evaluate FairSAM and baselines in terms of both robustness and fairness. The results show that FairSAM addresses the tension between robustness and fairness. FairSAM attains lower values on *Corrupted Degradation Disparity*, where lower is better.

Our contributions are as follows: 1) We identify and formalize the robustness bias challenge in image classification under data corruption and introduce *Corrupted Degradation Disparity* to evaluate performance degradation fairness across demographic subgroups. 2) We introduce **FairSAM**, a novel framework that integrates SAM with fairness-enhancing strategies to achieve both robustness and fairness in corrupted image classification tasks. 3) We validate FairSAM on multiple datasets and configurations, where it consistently improves accuracy and fairness compared with various baselines.

## 2 Related Work

### 2.1 Fairness in Machine Learning

Algorithmic fairness has become a central topic in machine learning as researchers increasingly recognize biases that disproportionately affect marginalized groups based on demographic factors such as gender, race, or age. In image classification, these biases can appear as unequal performance across subgroups, especially under challenging conditions such as image corruption. Despite its importance, fairness in the presence of image corruption remains an underexplored area. This gap matters because machine learning

models deployed in real-world environments frequently encounter corrupted data, which can amplify existing inequalities by disproportionately affecting certain demographic groups.

Existing research on fair machine learning focuses primarily on two objectives: (1) defining and identifying bias in machine learning models and (2) developing algorithms to effectively mitigate bias. Various fairness definitions have been proposed, with *statistical parity* being one of the most widely recognized. Statistical parity ensures that the likelihood of favorable outcomes remains similar across protected and non-protected groups. This can be quantified through metrics like *risk difference*, *risk ratio*, *relative change*, and *odds ratio* (Žliobaite, 2017). These metrics quantify fairness and support comparisons of bias in model predictions. Bias mitigation strategies fall into three main categories: pre-processing, in-processing, and post-processing methods. *Pre-processing* techniques modify the training data to remove potential biases before model training. Examples include *Massaging* (Kamiran & Calders, 2009), *reweighing* (Calders et al., 2009), and *Preferential Sampling* (Kamiran & Calders, 2012), which adjust data distributions to promote fairness. In contrast, *in-processing* methods (Calders & Verwer, 2010; Corbett-Davies et al., 2017; Kamiran et al., 2010; Zafar et al., 2017b; Wu et al., 2019; Dai et al., 2024; Jiang et al., 2023) introduce fairness constraints or regularization terms directly into the model objective so the learning algorithm optimizes fairness alongside accuracy. Finally, *post-processing* techniques (Awasthi et al., 2020; Hardt et al., 2016; Kamiran et al., 2012) adjust model predictions after training to correct for any biases detected in model outputs.

Despite significant progress, most of these methods have not specifically addressed fairness issues arising from image corruption, where different groups may experience varying degrees of accuracy loss. This gap motivates our simultaneous study of robustness and fairness under image corruption.

## 2.2 Fairness and Robustness

Foundational work in algorithmic fairness primarily focused on fairness across demographic subgroups, but real-world deployments have revealed the fragility of these guarantees. A model deemed "fair" in the laboratory can exhibit significant bias under natural variation in a production environment or deliberate manipulation by an adversary. This concern has expanded fairness research to include robustness to distribution shift and adversarial attacks.

**Fairness and Distribution Shift:** Distribution shift poses challenges to maintaining fairness across demographic groups. Sagawa et al. (2020) demonstrated that models can exhibit disparate performance across subgroups when the test distribution differs from the training distribution, particularly affecting minority groups. Koh et al. (2021) introduced the WILDS benchmark to systematically evaluate model performance under various types of distribution shift. The benchmark highlights how demographic subgroups can be disproportionately affected. Liu et al. (2021) showed that standard domain adaptation techniques can inadvertently amplify bias, while Zhao et al. (2024) proposed methods to maintain fairness guarantees under covariate shift.

**Fairness and Adversarial Attacks:** The vulnerability of fair models to adversarial perturbations has received considerable attention. Xu et al. (2021) demonstrated that adversarially trained models often exhibit increased bias against minority groups. This result reveals a fundamental tension between adversarial robustness and fairness. Sun et al. (2022) proposed adversarial training methods that explicitly account for fairness constraints, while Tran et al. (2022) showed that certain demographic groups are more susceptible to adversarial attacks than others. Mehrabi et al. (2021); Van et al. (2022); Solans et al. (2020) further explored how adversarial examples can target fairness and cause disproportionate performance degradation across subgroups. Nanda et al. (2021) showed that different demographic subgroups exhibit different levels of robustness, and that this disparity can produce unfairness.

Most current work emphasizes adversarial robustness rather than robustness to natural corruption, and few methods explicitly tackle the intersection of fairness and robustness under image corruption. We investigate this intersection and propose a method to mitigate such bias.

## 2.3 SAM and its Variants

Sharpness-Aware Minimization (Foret et al., 2021) was introduced to improve neural network generalization by identifying flatter minima in the loss landscape. SAM minimizes the maximum loss within a neighborhood around the current parameter setting rather than minimizing the loss at a single point. This approach yields solutions that are more resilient to small parameter perturbations and improves model generalization and robustness.

ImbSAM (Zhou et al., 2023) extends SAM's applicability to settings with extremely imbalanced data distributions, addressing the trade-off between sharpness and data imbalance. By incorporating strategies to handle imbalance, ImbSAM enhances generalization for certain long-tail classes and improves robustness in challenging training scenarios. Adaptive Sharpness-Aware Pruning (AdaSAP) (Bair et al., 2023) advances SAM's concept by focusing on pruning models to enhance both compactness and robustness. AdaSAP employs adaptive weight perturbations to regularize pruned models, improving their resilience against corrupted data while maintaining efficient model size. However, none of these SAM variants specifically target fairness across demographic subgroups, particularly under image corruption.

Our work advances SAM toward fairness and robustness to image corruption. We incorporate fairness mechanisms to preserve robust performance without sacrificing equity across sensitive demographic groups. This work addresses limitations in SAM and its variants by connecting robustness and fairness in corrupted image classification.

The foregoing lines of work address fairness under distribution shift or under adversarial perturbation, but not under *natural* corruption (e.g., sensor noise and weather conditions) that differs from both full distribution shift and adversarial attacks. Fairness under such natural corruption remains underexplored. We address this gap by introducing a metric and a method for equitable robustness under corruption.

## 2.4 Fairness in Generative Models

Recent fairness research on large-scale generative models has shifted from classical parity constraints toward distributional control of model outputs. For large language models, fairness concerns arise because pretraining data encode social stereotypes that can propagate to downstream behavior. Evaluation and mitigation methods therefore focus on both fine-tuning and prompting paradigms (Gallegos et al., 2024; Li et al., 2024). In text-to-image diffusion, Fair Diffusion (Friedrich et al., 2023) steers deployed models through fairness instructions without retraining, while Finetuning Text-to-Image Diffusion Models for Fairness (Shen et al., 2024) frames debiasing as distributional alignment over generated attributes. Balancing Act (Parihar et al., 2024) similarly uses distribution-guided sampling to match prescribed demographic distributions without full retraining. LightFair (Han et al., 2025) shows that lightweight debiasing of the text encoder can substantially reduce text-to-image bias with limited training and sampling overhead. These works suggest that traditional fair machine learning approaches may not apply directly to generative models. This setting requires fairness-aware optimization methods for large-scale models through parameter-efficient adaptation, where SAM-style perturbations and subgroup-aware reweighting may improve fairness robustness without full model retraining.

# 3 Preliminary

## 3.1 Fair Classification

We first formulate the fair classification problem under corrupted conditions. Consider a clean training dataset $\mathcal{D}_T = \left\{ (\mathbf{x}_i, \mathbf{y}_i, s_i) \right\}_{i=1}^{N}$ and a corrupted test dataset $\mathcal{D}_C = \left\{ (\mathbf{x}_i^c, \mathbf{y}_i, s_i) \right\}_{i=1}^{M}$, where $\mathbf{x}_i \in \mathcal{X}$ denotes an input image, $\mathbf{y}_i \in \mathcal{Y}$ denotes the ground-truth target, and $s_i \in \mathcal{S} = \{s^+, s^-\}$ represents a sensitive attribute. Here, $s^+$ and $s^-$ denote the advantaged and disadvantaged groups, respectively, and $\mathbf{x}_i^c$ denotes a corrupted input (e.g., an image with noise). The classification hypothesis space is $f(\mathbf{w}) : \mathcal{X} \to \mathcal{Y}$, parameterized by $\mathbf{w}$. Traditionally, fair classification seeks equal outcomes (e.g., demographic parity) or equal performance (e.g., equalized odds) across demographic subgroups.

## 3.2 Sharpness-Aware Minimization

Sharpness-Aware Minimization (SAM) is an optimization technique that enhances the generalization capability of neural networks by mitigating overfitting. Unlike traditional methods that solely minimize the loss at the current parameter values, SAM minimizes the maximum loss within a neighborhood around the current parameters. This strategy encourages the model to find "flatter" minima in the loss landscape, where surrounding regions exhibit uniformly low loss, improving generalization and robustness.

Consider a family of models parameterized by $\mathbf{w} \in \mathcal{W} \subseteq \mathbb{R}^d$, where $\mathcal{L}$ is the loss function and $\mathcal{D}_T$ is the training dataset. SAM minimizes an upper bound on the PAC-Bayesian generalization error. For a given $\rho > 0$ and norm $\| \cdot \|_p$ (typically $p = 2$), this bound is:

$$\mathcal{L}(\mathbf{w}) \leq \max_{\|\epsilon\|_p \leq \rho} \left[ \mathcal{L}_{\mathcal{D}_T}(\mathbf{w} + \epsilon) - \mathcal{L}_{\mathcal{D}_T}(\mathbf{w}) \right] + \mathcal{L}_{\mathcal{D}_T}(\mathbf{w}) + \frac{\lambda}{2} \|\mathbf{w}\|^2. \tag{1}$$

Therefore, the problem is a minimax problem:

$$\min_{\mathbf{w}} \max_{\|\boldsymbol{\epsilon}\|_p \leq \rho} \mathcal{L}_{\mathcal{D}_T}(\mathbf{w} + \boldsymbol{\epsilon}) + \frac{\lambda}{2} \|\mathbf{w}\|^2. \tag{2}$$

To solve this minimax problem, SAM performs the following iterative update at each iteration $t$:
First, compute the perturbation $\boldsymbol{\epsilon}_t$ as:

$$\boldsymbol{\epsilon}_t = \frac{\rho \cdot \text{sign}\left(\nabla \mathcal{L}_{\mathcal{D}_T}\left(\mathbf{w}_{t-1}\right)\right) \left|\nabla \mathcal{L}_{\mathcal{D}_T}\left(\mathbf{w}_{t-1}\right)\right|^{q-1}}{\left(\left\|\nabla \mathcal{L}_{\mathcal{D}_T}\left(\mathbf{w}_{t-1}\right)\right\|_q^q\right)^{1/p}} \tag{3}$$

where $1/p + 1/q = 1$, $\rho > 0$ is a hyperparameter controlling the neighborhood size, and $p$, $q$ denote norm parameters. The term $|\cdot|^{q-1}$ denotes the element-wise absolute value raised to the $(q-1)$ power. Typically, $p$ and $q$ are set to 2.

Second, update the model parameter $\mathbf{w}_t$ as:

$$\mathbf{w}_t = \mathbf{w}_{t-1} - \eta_t \left(\nabla \mathcal{L}_{\mathcal{D}_T}\left(\mathbf{w}_{t-1} + \boldsymbol{\epsilon}_t\right) + \lambda \mathbf{w}_{t-1}\right) \tag{4}$$

where $\lambda > 0$ is the parameter for weight decay, and $\eta_t > 0$ is the learning rate.

With $p = q = 2$, introducing an intermediate variable $\mathbf{u}_t$, we have:

$$\mathbf{u}_t = \mathbf{w}_{t-1} + \frac{\rho \nabla \mathcal{L}_{\mathcal{D}_T}\left(\mathbf{w}_{t-1}\right)}{\left\|\nabla \mathcal{L}_{\mathcal{D}_T}\left(\mathbf{w}_{t-1}\right)\right\|}, \tag{5}$$

$$\mathbf{w}_t = \mathbf{w}_{t-1} - \eta_t \left(\nabla \mathcal{L}_{\mathcal{D}_T}\left(\mathbf{u}_t\right) + \lambda \mathbf{w}_{t-1}\right). \tag{6}$$

By minimizing loss over a neighborhood rather than a single point, SAM finds parameter configurations that are less sensitive to small perturbations and that generalize better under corruption. This property of SAM motivates our development of FairSAM, which extends SAM to also address fairness concerns across demographic subgroups under corrupted conditions.

# 4 Proposed Method

## 4.1 New Fairness Notions for Corrupted Data

Traditional fair machine learning focuses primarily on performance disparity. However, performance degradation can be unevenly distributed across subgroups when data are corrupted. Our objective is to address this robustness-based fairness by training a model $f(\mathbf{w})$ that maintains consistent performance degradation across distinct subgroups, as measured by a specified metric. We formally define a new fairness metric for corrupted data as follows:

**Definition 1** (Corrupted Degradation Disparity)**.** Given a model $f$ trained on clean training data $\mathcal{D}_T$, we define *Corrupted Degradation* for a specific demographic group $s$ as:

$$\Delta p^s = |\mathbb{M}(\mathcal{D}_T^{S=s}, f) - \mathbb{M}(\mathcal{D}_C^{S=s}, f)|,$$

where $\mathbb{M}(\mathcal{D}_T^{S=s}, f)$ and $\mathbb{M}(\mathcal{D}_C^{S=s}, f)$ represent the performance metric values on clean training and corrupted test data, respectively, for subgroup $s$. We then define *Corrupted Degradation Disparity* as:

$$\Delta p = |\Delta p^{s^+} - \Delta p^{s^-}|.$$

$\square$

This definition quantifies how differently subgroups are impacted by data corruption. By measuring this disparity, we can identify subgroups that are disproportionately affected. Specifically, a smaller $\Delta p$ value indicates that the model's robustness is more equally distributed across subgroups. This metric can be naturally extended to multi-class classification tasks. We discuss the extension to multiple sensitive attributes and attributes with multiple values in Section 6.

### 4.2 FairSAM: Instance-Reweighted SAM

Sharpness-Aware Minimization (SAM) improves model robustness by encouraging "flat" minima in the loss landscape, where loss changes gradually with parameter variations. This property improves generalization and resilience to data corruption. However, SAM does not adequately address fairness as its robustness improvements are not uniformly distributed across demographic subgroups. In particular, SAM's accuracy gains can be unevenly allocated, and some disadvantaged subgroups can experience disproportionately high accuracy degradation.

To address this limitation, we introduce a reweighting mechanism that adjusts sample importance across subgroups so that SAM prioritizes both robustness and fairness. By allocating greater attention to samples from disadvantaged subgroups, this reweighting scheme balances gradient contributions from each group, mitigates robustness disparities, and ensures more equitable performance across demographic subgroups.

**From Vanilla SAM to FairSAM.** Consider a training dataset $\mathcal{D}_T = \left\{(\mathbf{x}_i, \mathbf{y}_i, s_i)\right\}_{i=1}^N$. Let $\ell_i(\mathbf{w})$ denote the loss for sample $i$ under model parameters $\mathbf{w}$. Vanilla SAM optimizes a per-instance perturbation objective:

$$\min_{\mathbf{w}} \frac{1}{n} \sum_{i=1}^n \max_{\|\epsilon_i\|_2 \leq \rho} \ell_i(\mathbf{w} + \epsilon_i), \tag{7}$$

where $\epsilon_i$ represents the adversarial perturbation for sample $i$ within a neighborhood of radius $\rho$. This un-weighted formulation allocates gradient contributions proportionally to group size. Majority groups dominate the optimization, which yields perturbations $\epsilon_i$ that vary across subgroups and produce unequal generalization.

**Fairness-Aware Reweighting.** We assign initial weights within each group as $\gamma_i = \frac{c}{n'}$, where $n'$ is the number of samples in the group containing the $i$-th sample and $c$ is the total sample weight assigned to that group. By assigning weights to samples of different groups, the classifier is encouraged to focus more on samples that are either misclassified or likely to be misclassified. Moreover, this approach ensures that the weighted representation of samples remains balanced across different groups. To equalize group influence regardless of size, we constrain weights to sum to a constant $c$ within each group:

$$\max_{\boldsymbol{\gamma}} \sum_s \sum_{i \in g_s} \gamma_i \max_{\|\epsilon_i\|_2 \leq \rho} \ell_i(\mathbf{w} + \epsilon_i) \qquad s.t. \quad \sum_{i \in g_s} \gamma_i = c, \quad \gamma_i \geq 0 \tag{8}$$

where $g_s$ collects the indices of samples belonging to the demographic group $s$. This constraint ensures that each demographic group $s$ contributes equally to the total objective, independent of its size $n_s$. The optimization in Equation 8 decomposes by group. For each group $s$, we solve:

$$\max_{\boldsymbol{\gamma}} \sum_{i=1}^{n'} \gamma_i \underbrace{\max_{\|\epsilon_i\|_2 \leq \rho} \ell_i(\mathbf{w} + \epsilon_i)}_{\text{per-sample SAM } \ell_s} \qquad s.t. \quad \boldsymbol{\gamma}^T 1 = c, \qquad \gamma_i \geq 0, \tag{9}$$

Within each group, samples with higher per-instance sharpness (larger $\max_{\|\epsilon_i\|_2 \leq \rho} \ell_i(\mathbf{w} + \epsilon_i)$) receive larger weights. This reweighting equalizes group influence and prioritizes harder examples within each group.

### 4.3 Efficient Computation via Per-Batch Perturbation

The formulation above assumes per-instance perturbations $\epsilon_i$, which require a separate adversarial perturbation for each sample. In practice, SAM computes a single per-batch perturbation $\boldsymbol{\epsilon}$ shared across all samples:

$$\min_{\mathbf{w}} \max_{\|\boldsymbol{\epsilon}\|_p \leq \rho} \frac{1}{n} \sum_{i=1}^{n} \ell_i(\mathbf{w} + \boldsymbol{\epsilon}). \tag{10}$$

This reduces computational cost from $n$ forward passes to one per batch. We adapt this efficiency to our fairness-aware objective by deriving a per-batch perturbation $\boldsymbol{\epsilon}$ that approximates the weighted per-instance formulation.

**Deriving the Fairness-Aware Perturbation.** We approximate each instance loss using its second-order Taylor expansion:

$$\ell_i(\mathbf{w} + \boldsymbol{\epsilon}) \approx \ell_i(\mathbf{w}) + \nabla \ell_i(\mathbf{w})^T \boldsymbol{\epsilon} + \frac{1}{2} \boldsymbol{\epsilon}^T H_i(\mathbf{w}) \boldsymbol{\epsilon}, \tag{11}$$

where $H_i(\mathbf{w})$ is the Hessian of $\ell_i$ at $\mathbf{w}$. Following standard approximations for sharpness-aware training, we assume a low-rank Hessian structure: $H_i(\mathbf{w}) = a_i \nabla \ell_{i,\boldsymbol{\gamma}}(\mathbf{w}) \nabla \ell_{i,\boldsymbol{\gamma}}(\mathbf{w})^T$ for $a_i > 0$. Under this assumption, the optimal per-instance perturbation direction aligns with the gradient $\nabla \ell_i(\mathbf{w})$, scaled by the instance sharpness $a_i$.

To combine fairness-aware reweighting with efficient per-batch computation, we construct a weighted batch loss and compute its gradient:

$$\begin{aligned}
\ell_b(\mathbf{w}) &= \sum_{i=1}^{N} g_i \ell_i(\mathbf{w}), \\
\boldsymbol{\epsilon}^* &= \rho \frac{\nabla \ell_b(\mathbf{w})}{\|\nabla \ell_b(\mathbf{w})\|_2},
\end{aligned} \tag{12}$$

where $g_i = a_i \|\nabla \ell_i(\mathbf{w})\|_2$ represents the combined influence of instance sharpness and gradient magnitude. The weights $g_i$ incorporate the fairness constraint through group membership and instance difficulty through sharpness. This formulation requires only one forward-backward pass per batch. It maintains SAM's computational efficiency while enforcing fairness across demographic groups.

## 5 Experiments

### 5.1 Datasets and Experiment Settings

We evaluate FairSAM on several widely used datasets, including CelebA (Liu et al., 2015), FairFace (Karkkainen & Joo, 2021), LFW (Huang et al., 2007), and CheXpert (Irvin et al., 2019), to measure robustness and fairness under corrupted data. Specifically, CelebA is imbalanced and has a substantial sample-size disparity between the advantaged and disadvantaged groups. This imbalance creates a challenging setting for subgroup fairness. Specifically, we select "Big Nose" and "Blond Hair" as target attributes and "Gender" and "Age" as sensitive attributes. In contrast, FairFace is balanced and has a roughly equal distribution of samples across demographic groups, which provides a controlled setting for evaluating FairSAM under balanced conditions. We choose "Age" as the sensitive attribute and "Gender" as the target attribute. For CheXpert, we follow the standard setting where the sensitive attribute is "Gender" and the target label is "Pleural Effusion". We use this setup for all CheXpert experiments. These datasets represent corrupted-data scenarios and support evaluation across diverse settings. To investigate FairSAM's fairness-aware generalization, we conduct out-of-distribution robustness experiments using the CelebA and LFW datasets, where models are trained on one dataset and tested on the other. All models are implemented in PyTorch and evaluated on an Ubuntu 20.04 LTS server with an Intel(R) Core(TM) i9-10900X CPU, 128 GB memory, and

an NVIDIA GeForce RTX 3070 GPU. Unless otherwise specified, we set the hyperparameters to $c = 1$ and $\rho = 0.05$. All source code is available in our repository[1].

| Methods | Test Data | Acc $s^+$ | $\Delta p^{s^+}$ | Acc $s^-$ | $\Delta p^{s^-}$ | Accuracy ↑ | $\Delta Acc$ ↓ | $\Delta p$ ↓ |
|---|---|---|---|---|---|---|---|---|
| Vanilla | clean | 0.8572 | 0.0115 | 0.7171 | 0.0862 | 0.8232 | 0.2148 | 0.0747 |
| | corrupted | 0.8457 | | 0.6309 | | **0.7901** | | |
| FairReg | clean | 0.6530 | 0.0215 | 0.6492 | 0.0588 | 0.6517 | **0.0411** | 0.0373 |
| | corrupted | 0.6315 | | 0.5904 | | 0.6217 | | |
| Reweighed | clean | 0.8527 | 0.0436 | 0.7156 | 0.0836 | 0.7983 | 0.1771 | 0.0400 |
| | corrupted | 0.8091 | | 0.6320 | | 0.7662 | | |
| GroupDRO | clean | 0.7979 | 0.0071 | 0.7144 | 0.0509 | 0.7722 | 0.1273 | 0.0438 |
| | corrupted | 0.7908 | | 0.6635 | | 0.7653 | | |
| SAM | clean | 0.8590 | 0.0090 | 0.7043 | 0.0666 | 0.8215 | 0.2123 | 0.0576 |
| | corrupted | 0.8500 | | 0.6377 | | 0.7984 | | |
| MSAM | clean | 0.8632 | 0.0128 | 0.6422 | 0.0756 | 0.8279 | 0.2082 | 0.0628 |
| | corrupted | 0.8504 | | 0.6422 | | 0.7997 | | |
| GroupSAM | clean | 0.8571 | 0.0074 | 0.7046 | 0.0534 | 0.8199 | 0.1985 | 0.0460 |
| | corrupted | 0.8497 | | 0.6512 | | 0.7809 | | |
| FairSAM (Ours) | clean | 0.8574 | 0.0399 | 0.7480 | 0.0499 | 0.8310 | 0.1194 | **0.0100** |
| | corrupted | 0.8175 | | 0.6981 | | 0.7885 | | |

Table 1: **Performance and fairness trade-off on CelebA.** The target attribute is "Big Nose", the sensitive attribute is "Age", and the corruption is level-3 snow noise. FairSAM achieves the lowest $\Delta p$ while preserving competitive corrupted accuracy.

## 5.2 Baseline Methods

All experiments are conducted using the ResNet-18 model architecture by default. For the DINOv3 backbone in Section 5.4, we use the ViT-S/16 distilled variant and adapt the corresponding baselines accordingly. We adapt ImbSAM (Zhou et al., 2023) to improve fairness-aware robustness by selectively applying SAM to the disadvantaged subgroup during training. We refer to this approach as group-specific SAM (**GroupSAM**). Specifically, we first reformulate Equation 2 as follows:

$$\min_{\mathbf{w}} \max_{\|\boldsymbol{\epsilon}\| \leq \rho} \left[ \mathcal{L}_{s^+}(\mathbf{w} + \boldsymbol{\epsilon}) + \mathcal{L}_{s^-}(\mathbf{w} + \boldsymbol{\epsilon}) \right] + \frac{\lambda}{2} \|\mathbf{w}\|^2, \tag{13}$$

where $\mathcal{L}_{s^+}$ and $\mathcal{L}_{s^-}$ represent the losses for the advantaged and disadvantaged groups, respectively. To improve fairness, we then apply SAM only to the disadvantaged group:

$$\min_{\mathbf{w}} \max_{\|\boldsymbol{\epsilon}^-\| \leq \rho} \mathcal{L}_{s^-}(\mathbf{w} + \boldsymbol{\epsilon}^-) + \mathcal{L}_{s^+}(\mathbf{w}) + \frac{\lambda}{2} \|\mathbf{w}\|^2 \tag{14}$$

where $\boldsymbol{\epsilon}^-$ is a perturbation specific to the disadvantaged group. The perturbation $\boldsymbol{\epsilon}^-$ in Equation 14 differs from $\boldsymbol{\epsilon}$ in Equation 13 because SAM is applied only to the disadvantaged group, so the perturbation is group-specific.

To make the sharpness-aware term explicit, we rewrite the objective as follows:

$$\min_{\mathbf{w}} \overbrace{\max_{\|\boldsymbol{\epsilon}^-\| \leq \rho} \left[ \mathcal{L}_{s^-}(\mathbf{w} + \boldsymbol{\epsilon}^-) - \mathcal{L}_{s^-}(\mathbf{w}) \right] + \mathcal{L}_{s^-}(\mathbf{w})}^{\text{Disadvantaged group-specific SAM term}}$$
$$+ \mathcal{L}_{s^+}(\mathbf{w}) + \frac{\lambda}{2} \|\mathbf{w}\|^2. \tag{15}$$

Additionally, we train several comparison models, including vanilla ResNet-18 (**Vanilla**), ResNet-18 with fairness regularizers (**FairReg**) (Zafar et al., 2017a), reweighed ResNet-18 (**Reweighed**) (Kamiran &

---

[1] http://tiny.cc/FairSAM

Calders, 2011), Group-specific SAM (**GroupSAM**), vanilla SAM (**SAM**), MSAM (Becker et al., 2025), GroupDRO (Sagawa et al., 2020), and FairSAM. We evaluate model performance on both clean and corrupted versions of the test data, focusing on accuracy across demographic subgroups (Young and Non-Young) under noise-free and noise-corrupted conditions. Following the noise settings from ImageNet-C (Hendrycks & Dietterich, 2019), we add several corruption types, including snow, Gaussian noise, and blur, at five severity levels (1 to 5).

| Methods | Test Data | Acc $s^+$ | $\Delta p^{s^+}$ | Acc $s^-$ | $\Delta p^{s^-}$ | Accuracy ↑ | $\Delta Acc$ ↓ | $\Delta p$ ↓ |
|---|---|---|---|---|---|---|---|---|
| Vanilla | clean | 0.9769 | 0.0006 | 0.9318 | 0.1083 | 0.9493 | 0.1528 | 0.1077 |
|  | corrupted | 0.9763 |  | 0.8235 |  | 0.8827 |  |  |
| FairReg | clean | 0.9442 | 0.0281 | 0.9532 | 0.1094 | 0.9387 | 0.1465 | 0.0813 |
|  | corrupted | 0.9723 |  | 0.8258 |  | 0.8824 |  |  |
| Reweighed | clean | 0.9627 | 0.0074 | 0.9351 | 0.1056 | 0.9457 | 0.1406 | 0.1130 |
|  | corrupted | 0.9701 |  | 0.8295 |  | 0.8839 |  |  |
| GroupDRO | clean | 0.9573 | 0.0441 | 0.9296 | 0.0244 | 0.9403 | **0.0080** | 0.0197 |
|  | corrupted | 0.9132 |  | 0.9052 |  | 0.9084 |  |  |
| SAM | clean | 0.9797 | 0.0168 | 0.9363 | 0.0575 | 0.9531 | 0.0841 | 0.0407 |
|  | corrupted | 0.9629 |  | 0.8788 |  | 0.9114 |  |  |
| MSAM | clean | 0.9801 | 0.0070 | 0.9372 | 0.0818 | 0.9539 | 0.1177 | 0.0748 |
|  | corrupted | 0.9731 |  | 0.8554 |  | 0.9010 |  |  |
| GroupSAM | clean | 0.9780 | 0.0094 | 0.9406 | 0.0468 | 0.9551 | 0.0748 | 0.0374 |
|  | corrupted | 0.9686 |  | 0.8938 |  | 0.9228 |  |  |
| FairSAM (Ours) | clean | 0.9734 | 0.0202 | 0.9412 | 0.0275 | 0.9570 | 0.0395 | **0.0073** |
|  | corrupted | 0.9532 |  | 0.9137 |  | **0.9291** |  |  |

Table 2: **Performance and fairness trade-off on CelebA.** The target attribute is "Blond Hair", the sensitive attribute is "Gender", and the corruption is level-3 Gaussian noise. FairSAM achieves the highest corrupted accuracy and the lowest $\Delta p$.

| Methods | Test Data | Acc $s^+$ | $\Delta p^{s^+}$ | Acc $s^-$ | $\Delta p^{s^-}$ | Accuracy ↑ | $\Delta Acc$ ↓ | $\Delta p$ ↓ |
|---|---|---|---|---|---|---|---|---|
| Vanilla | clean | 0.7396 | 0.1704 | 0.8296 | 0.2008 | 0.7800 | 0.0596 | 0.0304 |
|  | corrupted | 0.5692 |  | 0.6288 |  | 0.5961 |  |  |
| GroupDRO | clean | 0.7330 | 0.1838 | 0.7965 | 0.3285 | 0.7618 | 0.0812 | 0.1447 |
|  | corrupted | 0.5492 |  | 0.4680 |  | 0.5127 |  |  |
| SAM | clean | 0.7727 | 0.1518 | 0.8781 | 0.1732 | 0.8201 | 0.0840 | 0.0214 |
|  | corrupted | 0.6209 |  | 0.7049 |  | 0.6885 |  |  |
| MSAM | clean | 0.7549 | 0.1470 | 0.8359 | 0.1886 | 0.7914 | **0.0394** | 0.0416 |
|  | corrupted | 0.6079 |  | 0.6473 |  | 0.6253 |  |  |
| GroupSAM | clean | 0.7550 | 0.1253 | 0.8333 | 0.1441 | 0.7904 | 0.0595 | 0.0188 |
|  | corrupted | 0.6297 |  | 0.6892 |  | 0.6563 |  |  |
| FairSAM | clean | 0.7837 | 0.1467 | 0.9075 | 0.1539 | 0.8394 | 0.1166 | **0.0072** |
|  | corrupted | 0.6370 |  | 0.7536 |  | **0.6893** |  |  |

Table 3: **Performance and fairness trade-off on FairFace.** The target attribute is "Gender", the sensitive attribute is "Age", and the corruption is level-5 Gaussian noise. FairSAM achieves the highest corrupted accuracy and the lowest $\Delta p$.

### 5.3 Trade-off between Fairness and Performance

We investigate robustness disparity across subgroups on both balanced and imbalanced datasets with different sensitive and target attributes. We train models using the clean training data and evaluate those models on noisy data. Tables 1, 2, and 3 compare methods in terms of accuracy, corrupted degradation disparity, and performance disparity. Values in bold indicate the **best** performance among all methods, while underlined values indicate the second-best performance. We run all experiments *three times* and report average accuracy. We omit standard deviations because training is stable, with standard deviations usually below 0.1%.

| Methods | Test Data | Acc $s^+$ | $\Delta p^{s^+}$ | Acc $s^-$ | $\Delta p^{s^-}$ | Accuracy ↑ | $\Delta Acc$ ↓ | $\Delta p$ ↓ |
|---|---|---|---|---|---|---|---|---|
| Vanilla | clean | 0.8095 | 0.0265 | 0.8188 | 0.1803 | 0.8162 | 0.1445 | 0.1538 |
| | corrupted | 0.7830 | | 0.6385 | | 0.7179 | | |
| GroupDRO | clean | 0.8163 | 0.1020 | 0.7964 | 0.0020 | 0.8119 | 0.0841 | 0.1000 |
| | corrupted | 0.7143 | | 0.7984 | | 0.7606 | | |
| SAM | clean | 0.8067 | 0.0508 | 0.8547 | 0.0448 | 0.8333 | 0.0540 | 0.0060 |
| | corrupted | 0.7559 | | 0.8099 | | 0.7863 | | |
| MSAM | clean | 0.8088 | 0.1302 | 0.8096 | 0.0231 | 0.8112 | 0.1079 | 0.1071 |
| | corrupted | 0.6786 | | 0.7865 | | 0.7307 | | |
| GroupSAM | clean | 0.7722 | 0.0694 | 0.8258 | 0.0493 | 0.8034 | 0.0737 | 0.0201 |
| | corrupted | 0.7028 | | 0.7765 | | 0.7436 | | |
| FairSAM (Ours) | clean | 0.8127 | 0.0114 | 0.8144 | 0.0160 | 0.8162 | **0.0029** | **0.0046** |
| | corrupted | **0.8013** | | 0.7984 | | **0.8034** | | |

Table 4: **Performance and fairness trade-off on CheXpert.** The target label is "Pleural Effusion", and the sensitive attribute is "Gender". FairSAM achieves the lowest $\Delta p$ and $\Delta Acc$ and the highest corrupted accuracy.

**Corrupted Degradation Disparity**. The degradation disparity results are reported in the $\Delta p$ column in Tables 1, 2, and 3. FairSAM consistently achieves the best balance of fairness and accuracy among the baselines. Specifically, FairSAM outperforms common fairness-aware methods, such as FairReg and Reweighed, which improve subgroup fairness but suffer from notable accuracy degradation. In contrast, SAM and GroupSAM show strong generalization and robustness against corrupted data but retain higher performance-degradation disparities across demographic subgroups.

To further validate FairSAM, we conduct experiments on FairFace, a balanced dataset designed for fairness assessment. As shown in Table 3, the FairFace results are consistent with the trends observed on CelebA. FairSAM achieves the lowest corrupted degradation disparity among all methods. FairSAM also attains the highest accuracy through its balanced treatment of samples from both advantaged and disadvantaged groups. This result shows that FairSAM maintains robust performance and fairness even in datasets with balanced demographic distributions.

**Performance Disparity**. The $\Delta Acc$ column measures performance disparity among subgroups on corrupted images. Tables 1 and 2 show that FairSAM improves fairness across diverse target and sensitive attributes while maintaining consistent performance. Although FairReg achieves a notable fairness level, this improvement comes at a substantial cost to the performance of the advantaged group, resulting in reduced overall accuracy. FairSAM balances fairness and accuracy and yields equitable outcomes without compromising the performance of any subgroup.

We extend our evaluation to the CheXpert medical imaging dataset to assess FairSAM's generalizability beyond facial recognition. Table 4 reports performance on CheXpert across gender subgroups. FairSAM achieves the lowest corrupted degradation disparity ($\Delta p = 0.0046$) and performance disparity ($\Delta Acc = 0.0029$) among all methods. FairSAM also attains the highest corrupted accuracy (0.8034) with low subgroup disparity. These results confirm that FairSAM generalizes to medical imaging while preserving fairness.

### 5.4 Performance on Asymmetric Noise Levels and Different Backbones

We evaluate model robustness under asymmetric noise corruption across two backbone architectures. Table 5 reports results with DINOv3, and Table 6 reports results with ResNet-18. Both settings apply level-4 noise to subgroup $s^+$ and level-5 noise to $s^-$. This evaluation tests whether models maintain fairness when the privileged group faces less severe corruption than the disadvantaged group. Regarding degradation disparity $\Delta p$ and performance disparity $\Delta Acc$, FairSAM achieves strong fairness metrics across both settings. On DINOv3, FairSAM achieves the lowest degradation disparity ($\Delta p = 0.0678$) and performance disparity ($\Delta Acc = 0.1745$), outperforming SAM ($\Delta p = 0.1122$, $\Delta Acc = 0.1870$) and GroupDRO ($\Delta p = 0.1187$, $\Delta Acc = 0.1932$). On ResNet-18, FairSAM achieves the lowest degradation disparity ($\Delta p = 0.0694$) and the

highest corrupted overall accuracy (0.7334). Its performance disparity is $\Delta Acc = 0.0544$, which improves over GroupDRO (0.1725) but is higher than SAM (0.0194), MSAM (0.0358), and GroupSAM (0.0032). FairSAM also obtains the highest corrupted accuracy for both subgroups. These results show that FairSAM consistently reduces degradation disparity while preserving strong corrupted accuracy across backbone architectures and noise asymmetries.

| Methods | Test Data | Acc $s^+$ | $\Delta p^{s^+}$ | Acc $s^-$ | $\Delta p^{s^-}$ | Accuracy ↑ | $\Delta Acc$ ↓ | $\Delta p$ ↓ |
|---------|-----------|-----------|------------------|-----------|------------------|------------|----------------|--------------|
| Vanilla | clean | 0.8493 | 0.1935 | 0.9235 | 0.0795 | 0.8828 | 0.1882 | 0.1140 |
|         | corrupted | 0.6558 | | 0.8440 | | 0.7396 | | |
| GroupDRO | clean | 0.8506 | 0.2026 | 0.9251 | 0.0839 | 0.8841 | 0.1932 | 0.1187 |
|          | corrupted | 0.6480 | | 0.8412 | | 0.7342 | | |
| SAM | clean | 0.8522 | 0.1910 | 0.9270 | 0.0788 | 0.8858 | 0.1870 | 0.1122 |
|     | corrupted | 0.6612 | | 0.8482 | | 0.7445 | | |
| MSAM | clean | 0.8547 | 0.1969 | 0.9238 | 0.0790 | 0.8857 | 0.1870 | 0.1179 |
|      | corrupted | 0.6578 | | 0.8448 | | 0.7410 | | |
| GroupSAM | clean | 0.8508 | 0.1888 | 0.9302 | 0.0737 | 0.8866 | 0.1945 | 0.1151 |
|          | corrupted | 0.6620 | | 0.8565 | | 0.7486 | | |
| FairSAM (Ours) | clean | 0.8334 | 0.1629 | 0.9401 | 0.0951 | 0.8815 | **0.1745** | **0.0678** |
|                | corrupted | 0.6705 | | 0.8450 | | **0.7484** | | |

Table 5: **Performance and fairness trade-off on FairFace under asymmetric noise.** The backbone is DINOv3. The corruption is level-4 Gaussian noise for $s^+$ and level-5 Gaussian noise for $s^-$. FairSAM achieves the lowest $\Delta p$ and $\Delta Acc$ under this backbone.

| Methods | Test Data | Acc $s^+$ | $\Delta p^{s^+}$ | Acc $s^-$ | $\Delta p^{s^-}$ | Accuracy ↑ | $\Delta Acc$ ↓ | $\Delta p$ ↓ |
|---------|-----------|-----------|------------------|-----------|------------------|------------|----------------|--------------|
| Vanilla | clean | 0.7396 | 0.1089 | 0.8296 | 0.2008 | 0.7800 | **0.0019** | 0.0919 |
|         | corrupted | 0.6307 | | 0.6288 | | 0.6290 | | |
| GroupDRO | clean | 0.7330 | 0.0925 | 0.7965 | 0.3285 | 0.7618 | 0.1725 | 0.2360 |
|          | corrupted | 0.6405 | | 0.4680 | | 0.5633 | | |
| SAM | clean | 0.7727 | 0.0872 | 0.8781 | 0.1732 | 0.8201 | 0.0194 | 0.0860 |
|     | corrupted | 0.6855 | | 0.7049 | | 0.6905 | | |
| MSAM | clean | 0.7549 | 0.0718 | 0.8359 | 0.1886 | 0.7914 | 0.0358 | 0.1168 |
|      | corrupted | 0.6831 | | 0.6473 | | 0.6838 | | |
| GroupSAM | clean | 0.7550 | 0.0626 | 0.8333 | 0.1441 | 0.7904 | 0.0032 | 0.0815 |
|          | corrupted | 0.6924 | | 0.6892 | | 0.6983 | | |
| FairSAM (Ours) | clean | 0.7837 | 0.0845 | 0.9075 | 0.1539 | 0.8394 | 0.0544 | **0.0694** |
|                | corrupted | **0.6992** | | **0.7536** | | **0.7334** | | |

Table 6: **Performance and fairness trade-off on FairFace under asymmetric noise.** The backbone is ResNet-18. The corruption is level-4 Gaussian noise for $s^+$ and level-5 Gaussian noise for $s^-$. FairSAM achieves the lowest $\Delta p$ and the highest corrupted accuracy.

## 5.5 Ablation Study

We conduct an ablation study across multiple noise levels to thoroughly evaluate FairSAM's robustness and fairness. This study tests whether FairSAM consistently maintains strong accuracy and fairness across image corruption severities and compares it with SAM-based baselines. Specifically, we introduce incremental levels of snow noise, ranging from mild (level 1) to severe (level 5), to the test datasets. For each noise level, we measure accuracy and *Corrupted Degradation Disparity* across all methods to assess the balance between robustness and fairness. Figure 1 shows that FairSAM consistently outperforms all baselines in fairness across noise levels. FairSAM maintains the lowest $\Delta p$ at every noise level with comparable accuracy, which indicates a better fairness-accuracy trade-off across subgroups under various corruption conditions.

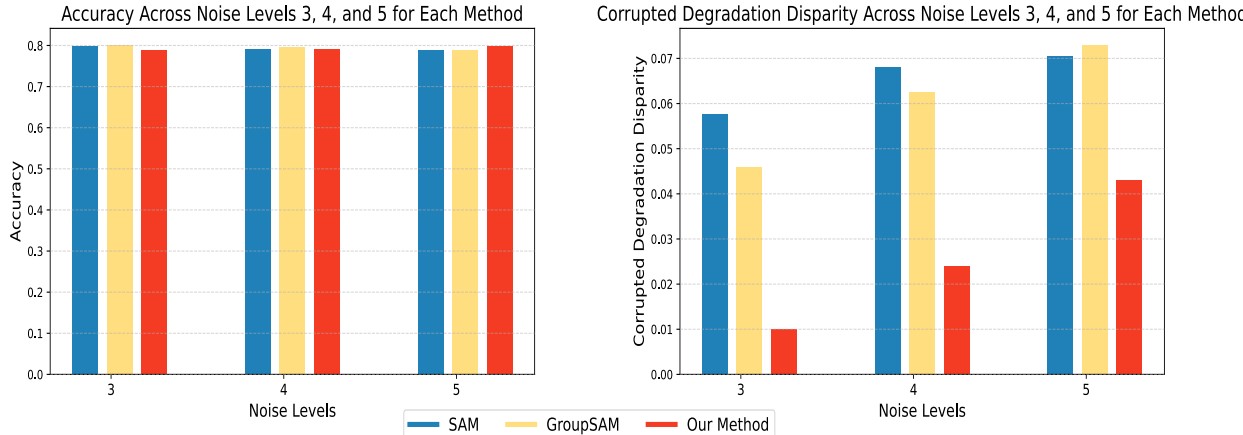

Figure 1: **Accuracy and Corrupted Degradation Disparity across snow noise levels.** The figure compares SAM-based methods under increasing snow corruption. FairSAM maintains the lowest $\Delta p$ with comparable accuracy.

## 5.6 Hyperparameter Sensitivity

We evaluate FairSAM's hyperparameter sensitivity to the perturbation radius $\rho$ and the per-group weight budget $c$ on FairFace. The perturbation radius controls the strength of penalizing sharp minima used for the sharpness-aware update. Table 7 reports a grid search over $\rho$ from 0.01 to 0.10. The default value $\rho = 0.05$ yields the lowest mean corrupted degradation disparity ($\Delta p = 0.0069 \pm 0.0054$) and preserves balanced corrupted subgroup accuracy. Smaller $\rho$ values preserve clean accuracy, but they leave larger degradation disparities. Larger values, especially $\rho = 0.08$-$0.10$, reduce clean accuracy and do not improve degradation fairness. These results support $\rho = 0.05$ as a balanced setting for robustness, fairness, and accuracy.

| $\rho$ | Test Data | Acc $s^+$ | Acc $s^-$ | $\Delta p \downarrow$ |
|---|---|---|---|---|
| 0.01 | clean | $0.7824 \pm 0.0041$ | $0.8978 \pm 0.0097$ | $0.0183 \pm 0.0144$ |
| | corrupted | $0.6332 \pm 0.0321$ | $0.7303 \pm 0.0289$ | |
| 0.02 | clean | $0.7856 \pm 0.0020$ | $0.9049 \pm 0.0056$ | $0.0113 \pm 0.0276$ |
| | corrupted | $0.6401 \pm 0.0261$ | $0.7481 \pm 0.0273$ | |
| 0.03 | clean | $0.7849 \pm 0.0040$ | $0.9090 \pm 0.0014$ | $0.0253 \pm 0.0086$ |
| | corrupted | $0.6499 \pm 0.0071$ | $0.7487 \pm 0.0168$ | |
| 0.04 | clean | $0.7827 \pm 0.0024$ | $0.9074 \pm 0.0032$ | $0.0176 \pm 0.0053$ |
| | corrupted | $0.6376 \pm 0.0073$ | $0.7447 \pm 0.0100$ | |
| 0.05 | clean | $0.7762 \pm 0.0035$ | $0.9003 \pm 0.0037$ | $\mathbf{0.0069 \pm 0.0054}$ |
| | corrupted | $0.6466 \pm 0.0049$ | $0.7638 \pm 0.0112$ | |
| 0.06 | clean | $0.7764 \pm 0.0036$ | $0.8867 \pm 0.0329$ | $0.0157 \pm 0.0097$ |
| | corrupted | $0.6503 \pm 0.0046$ | $0.7449 \pm 0.0240$ | |
| 0.07 | clean | $0.7678 \pm 0.0067$ | $0.8917 \pm 0.0042$ | $0.0099 \pm 0.0075$ |
| | corrupted | $0.6321 \pm 0.0048$ | $0.7461 \pm 0.0072$ | |
| 0.08 | clean | $0.7574 \pm 0.0069$ | $0.8824 \pm 0.0064$ | $0.0097 \pm 0.0105$ |
| | corrupted | $0.6281 \pm 0.0072$ | $0.7434 \pm 0.0091$ | |
| 0.09 | clean | $0.7368 \pm 0.0108$ | $0.8605 \pm 0.0057$ | $0.0114 \pm 0.0081$ |
| | corrupted | $0.6231 \pm 0.0077$ | $0.7354 \pm 0.0066$ | |
| 0.10 | clean | $0.7145 \pm 0.0161$ | $0.8374 \pm 0.0130$ | $0.0139 \pm 0.0049$ |
| | corrupted | $0.6211 \pm 0.0070$ | $0.7301 \pm 0.0063$ | |

Table 7: **Sensitivity analysis of $\rho$ on FairFace.** The target attribute is "Gender", the sensitive attribute is "Age", and the corruption is Gaussian noise for both Young and Non-Young groups. Results are mean $\pm$ standard deviation.

Table 8 reports a grid search over the per-group weight budget $c$. FairSAM constrains the sample weights in each demographic group to sum to $c$, so each group contributes equally to the weighted objective regardless of group size. The default value $c = 1$ achieves the lowest mean degradation disparity ($\Delta p = 0.0069 \pm 0.0054$) and the highest corrupted accuracies for both subgroups among the tested values. A smaller value ($c = 0.5$) lowers corrupted accuracy while larger values ($c = 1.5$ and $c = 2$) reduce clean and corrupted accuracies. Together, the two sensitivity studies validate the default setting $\rho = 0.05$ and $c = 1$ is appropriate for FairSAM.

| $c$ | Test Data | Acc $s^+$ | Acc $s^-$ | $\Delta p \downarrow$ |
|---|---|---|---|---|
| 0.5 | clean | $0.7821 \pm 0.0014$ | $0.9014 \pm 0.0007$ | $0.0085 \pm 0.0044$ |
| | corrupted | $0.6266 \pm 0.0217$ | $0.7374 \pm 0.0240$ | |
| 1 | clean | $0.7762 \pm 0.0035$ | $0.9003 \pm 0.0037$ | $\mathbf{0.0069 \pm 0.0054}$ |
| | corrupted | $0.6466 \pm 0.0049$ | $0.7638 \pm 0.0112$ | |
| 1.5 | clean | $0.6953 \pm 0.0045$ | $0.8222 \pm 0.0062$ | $0.0188 \pm 0.0071$ |
| | corrupted | $0.6246 \pm 0.0052$ | $0.7327 \pm 0.0118$ | |
| 2 | clean | $0.7227 \pm 0.0037$ | $0.8481 \pm 0.0049$ | $0.0123 \pm 0.0084$ |
| | corrupted | $0.6062 \pm 0.0061$ | $0.7193 \pm 0.0134$ | |

Table 8: **Sensitivity analysis of $c$ on FairFace.** The target attribute is "Gender", the sensitive attribute is "Age", and the corruption is Gaussian noise for both Young and Non-Young groups. Results are mean $\pm$ standard deviation.

| Method | Train $\rightarrow$ Test | Acc $s^+$ | Acc $s^-$ | $\Delta p \downarrow$ |
|---|---|---|---|---|
| SAM | CelebA $\rightarrow$ CelebA | 0.8590 | 0.7043 | 0.0210 |
| | CelebA $\rightarrow$ LFW | 0.5946 | 0.4189 | |
| GroupSAM | CelebA $\rightarrow$ CelebA | 0.8570 | 0.7042 | 0.1865 |
| | CelebA $\rightarrow$ LFW | 0.5363 | 0.5700 | |
| FairSAM | CelebA $\rightarrow$ CelebA | 0.8575 | 0.7480 | **0.0188** |
| | CelebA $\rightarrow$ LFW | 0.5341 | 0.4058 | |

Table 9: **Out-of-distribution performance from CelebA to LFW.** The target attribute is "Big Nose", and the sensitive attribute is "Age". FairSAM achieves the lowest $\Delta p$ across the evaluated methods.

| Method | Train $\rightarrow$ Test | Acc $s^+$ | Acc $s^-$ | $\Delta p \downarrow$ |
|---|---|---|---|---|
| SAM | LFW $\rightarrow$ LFW | 0.7714 | 0.7687 | 0.1456 |
| | LFW $\rightarrow$ CelebA | 0.6863 | 0.5380 | |
| GroupSAM | LFW $\rightarrow$ LFW | 0.7784 | 0.7963 | 0.1499 |
| | LFW $\rightarrow$ CelebA | 0.6590 | 0.5270 | |
| FairSAM | LFW $\rightarrow$ LFW | 0.7893 | 0.7984 | **0.1066** |
| | LFW $\rightarrow$ CelebA | 0.6668 | 0.5511 | |

Table 10: **Out-of-distribution performance from LFW to CelebA.** The target attribute is "Big Nose", and the sensitive attribute is "Age". FairSAM achieves the lowest $\Delta p$ across the evaluated methods.

## 5.7 Out-of-distribution Generalization

To further assess fairness-aware generalization, we conduct an out-of-distribution experiment. This evaluation measures the performance degradation difference between in-distribution and out-of-distribution test data for each demographic subgroup, using a metric similar to *Corrupted Degradation Disparity*. This approach estimates model robustness and fairness across datasets. As shown in Tables 9 and 10, FairSAM consistently shows the lowest $\Delta p$ among the evaluated methods. In contrast, GroupSAM, by disregarding the loss landscape flatness for the advantaged group, risks shifting this group into a disadvantaged position, potentially creating new imbalances.

## 6 Discussion

Our work reveals several insights about the intersection of robustness and fairness in machine learning systems. First, traditional robustness methods improve overall performance under corruption but can inadvertently worsen fairness disparities across demographic subgroups. This gap is a blind spot in robustness research because aggregate metrics can mask inequities. Second, fairness-aware optimization during training improves equitable outcomes without large sacrifices in overall accuracy. FairSAM's balanced perturbation strategy shows that incorporating fairness into the optimization is feasible and useful for robust systems. Third, the benefits of fairness-aware robustness extend beyond imbalanced datasets to balanced ones. This result indicates that demographic disparities in model performance under corruption are not solely attributable to class imbalance but reflect algorithmic biases that require targeted intervention.

The implications of this work extend beyond technical contributions to machine learning. As AI systems are increasingly deployed in high-stakes applications such as healthcare, criminal justice, and employment screening, these systems must maintain fairness under real-world conditions. This work addresses a gap in AI safety research: robustness and fairness are interconnected and must be addressed jointly. Robust models often exhibit disparate performance degradation across demographic groups. If unaddressed, this disparity can perpetuate or amplify societal inequalities.

**Limitations and future work.** FairSAM yields strong results, but several limitations remain. Our current evaluation focuses primarily on image classification tasks with binary sensitive attributes. For multiple sensitive attributes, let $\mathbf{S} = (S_1, \ldots, S_m)$ denote the joint sensitive-attribute tuple, and let $\mathcal{S}$ denote the set of all intersectional groups. For each group $\mathbf{s} \in \mathcal{S}$, corruption degradation can be written as:

$$\Delta p^{\mathbf{s}} = \left| \mathbb{M}(\mathcal{D}_T^{\mathbf{S}=\mathbf{s}}, f) - \mathbb{M}(\mathcal{D}_C^{\mathbf{S}=\mathbf{s}}, f) \right|, \tag{16}$$

where $\mathcal{D}_T^{\mathbf{S}=\mathbf{s}}$ and $\mathcal{D}_C^{\mathbf{S}=\mathbf{s}}$ denote clean and corrupted data for group $\mathbf{s}$. With multiple groups, a worst-case pairwise disparity can measure the largest robustness gap across intersectional groups:

$$\Delta p^{\mathrm{multi}} = \max_{\mathbf{s}, \mathbf{s}' \in \mathcal{S}} \left| \Delta p^{\mathbf{s}} - \Delta p^{\mathbf{s}'} \right|. \tag{17}$$

A smaller $\Delta p^{\mathrm{multi}}$ indicates more equitable robustness across intersectional groups. Future work should evaluate this extension across multiple sensitive attributes and other domains (Kong, 2022; Foulds et al., 2020; Pastor & Bonchi, 2024). Additionally, our fairness metrics represent one perspective on measuring equitable performance degradation. Alternative fairness criteria (Hardt et al., 2016; McNamara, 2019; Awasthi et al., 2020) and their integration into robust optimization frameworks remain important directions for future research.

## 7 Conclusion

We address the dual challenges of fairness and robustness in corrupted image classification. We introduce a novel metric for assessing unequal performance degradation in corrupted environments. We further develop FairSAM, a framework that couples robustness and fairness to support equitable performance across demographic subgroups. Experiments across multiple datasets and corruption conditions show that FairSAM consistently improves the trade-off between fairness and performance over baseline methods. FairSAM maintains robust performance and fairness across subgroups and supports resilient machine learning in real-world applications. Future work will explore additional corruption scenarios and extend FairSAM to broader fair and robust image classification settings.

## Acknowledgments

This work was supported in part by NSF 2242812, 2520496, and SC EPSCoR 24-GA02.

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
