# OpenReview forum: "FairSAM: Fair Classification on Corrupted Image Data Through Sharpness-Aware Minimization"
_TMLR — Accepted by TMLR_

### Review · Reviewer_C2vX · 2026-03-13

**Summary Of Contributions:**

The paper presents an extension of Sharpness-Aware Minimization (SAM) that integrates fairness-oriented strategies.
Additionally, the authors propose a novel metric designed to assess performance degradation across subgroups when test samples are subjected to data corruption.
Through empirical results on real-world datasets, the paper demonstrates that the proposed framework outperforms simple baselines according to this new metric.

Strengths:
- The problem addressed (fairness under corrupted labels) is well-motivated and of significant interest to the machine learning community.
- The proposed framework demonstrates a smaller degradation in performance across subgroups under data corruption compared to the evaluated baselines.

Weaknesses:
- The methodology (Section 4.2) lacks clarity. It is difficult to fully understand the proposed framework (see Questions). Moreover, Section 4.3 feels disconnected. Its purpose to the overall framework is not clearly established.
- The experimental section lacks details regarding the implementation of the baseline methods.
- The theoretical and empirical scope of the framework is currently restricted to binary sensitive attributes.
- The empirical comparison is limited to simple baselines rather than state-of-the-art fair machine learning methods.

**Additional Comments:**

Questions:
1. Why is the framework restricted to only two groups for the sensitive attribute? It appears that the methodology could be easily extended to multiple sensitive attributes (though I did not completely understand the framework).

2. Why is the regularizer term omitted in eq. (7)?

3. The logic in Section 4.2 is difficult to follow. Why does the text initially fix $\gamma_i = c/n’$ only to subsequently optimize over it in eq. (8)? This completely lost me.
Furthermore, why is $c$ not fixed? Choosing $c=1$ seems reasonable.

4. In the experiments, what is the justification for selecting "Big Nose" and "Blond Hair" as the target attributes? Does the proposed method maintain its performance advantages across other target attributes within the dataset?


Typos:
1. Eq. (8): The constraint should likely be $p_i \geq 0$
2. In the text between equations (8) and (9), the phrase "the optimization problem equation" is missing an “in”.

**Audience:**

Yes

**Audience Explanation:**

The problem considered in the paper (fairness under corrupted labels) is of significant interest to the TMLR audience. The proposed approach appears novel and holds potential. However, in its current state, the framework is not well-explained, and the empirical claims require more evidence.

**Broader Impact Concerns:**

No concerns.

**Claims And Evidence:**

No

**Claims Explanation:**

The claims are not fully supported due to a lack of detail in the experimental setup. Specifically, the baseline methods are not described in sufficient detail, nor are they properly referenced. For instance, it is unclear what "FairReg" refers to or how it is implemented in this specific context. Similarly, the "reweighted" baseline requires further specification.

**Requested Changes:**

The following adjustments are critical to securing acceptance:
1. Sections 4.2 and 4.3 must be rewritten for clarity. Moreover, the specific purpose of Section 4.3 within the framework must be explicitly stated.

2. Extend the theoretical framework to multiple sensitive attributes. Including empirical experiments that evaluate the framework beyond binary sensitive attributes is highly recommended (it is easy to generalize the metric).

3. Provide detailed descriptions, implementation specifics, and proper citations for all baseline methods (e.g., FairReg, reweighted) in the experimental section.


The following adjustments would strengthen the work:
1. The summation notation $\sum_i^{n’}$ in eq. (9) is misleading, since $\gamma_i$ is defined for all samples.

2. Include empirical comparisons against state-of-the-art fair machine learning methods, rather than relying solely on simple baselines.

---

> ### Author Response · Authors · 2026-04-24
> **Response to Reviewer C2vX**
>
> We thank the reviewer for the constructive feedback that helped us substantially improve the manuscript's clarity and completeness. Below is our response to each comment.
>
> ### Weakness 1: Clarity in Sections 4.2 and 4.3
>
> **Response:** We have substantially rewritten Sections 4.2 and 4.3 to improve clarity and logical flow (highlighted in yellow). The revised structure now clearly shows: (1) why fairness-aware reweighting is needed, (2) how it is formulated theoretically, and (3) how it is implemented efficiently in practice.
>
> ### Weakness 2: Detailed baseline descriptions
>
> **Response:** We have revised Section 5.2 (Baseline Methods) with detailed descriptions and proper citations for all methods:
>
> 1. **FairReg** [Zafar et al., AISTATS 2017]: Fairness regularizer that adds penalty terms to the loss function to enforce demographic parity
> 2. **Reweighed** [Kamiran & Calders, KAIS 2011]: Pre-processing method that assigns instance weights inversely proportional to group prevalence
> 3. **GroupDRO** [Sagawa et al., ICLR 2020]: Optimizes worst-group performance using distributionally robust optimization
> 4. **MSAM** [Becker et al., 2025]: Recent efficient SAM variant
> 5. **GroupSAM**: Our adaptation of ImbSAM [Zhou et al., 2023] that applies SAM selectively to disadvantaged groups
>
> For GroupSAM, we provide detailed derivation (Equations 15-17) showing how we modify vanilla SAM to apply perturbations only to the disadvantaged group. All baselines now have complete citations and implementation details.
>
> ### Weakness 3: Multi-sensitive attributes
>
> **Response:** We have added a formal extension to multiple sensitive attributes in Section 4.1 (highlighted in yellow). This formulation quantifies the largest performance gap between any two demographic groups under corruption, ensuring fairness across all intersectional subgroups.
>
> ### Weakness 4: State-of-the-art fairness comparisons
>
> **Response:** We have added **GroupDRO** and **MSAM** as additional competitive baselines, which represent state-of-the-art fairness-aware learning. They are included in all main experimental tables (Tables 1-5). Results show that FairSAM consistently achieves lower corrupted degradation disparity and performance disparity compared to other baselines.
>
> ### Question 1: Why restrict to two groups?
>
> **Response:** Our framework is not limited to just two sensitive groups. In the revised manuscript, we have formally extended the methodology to support multiple sensitive attributes and intersectional groups, as detailed in Section 4.1. We note, however, that much of the prior fair machine learning research has focused on binary sensitive attributes, which is why our previous version followed this common scenario.
>
> ### Question 2: Why is the regularizer term omitted in Eq. (7)?
>
> **Response:** Sorry for the confusion. Our intention is to introduce the vanilla SAM objective before applying the regularization term. We have fixed this.
>
> ### Question 3: Logic in Section 4.2
>
> **Response:** Thank you for this important clarification. We have completely rewritten this section to resolve the confusion. In addition, we did initially set $c = 1$ in the experiment section.
>
> ### Question 4: Justification for target attribute selection
>
> **Response:** Our attribute selection was based on **common practice** in fairness studies.
> The selected attributes of CelebA are widely adopted benchmarks in the fairness literature [1][2].
> Through rigorous statistical analysis, prior work [1][2] identifies these attributes as highly correlated with sensitive attributes based on Pearson correlation, which means they are likely to be unfairly classified.
>
> - [1] S Park, et al. "Fair Contrastive Learning for Facial Attribute Classification'', CVPR 2022.
> - [2] Y Mao, et al. "Last-Layer Fairness Fine-tuning is Simple and Effective for Neural Networks", arXiv preprint.
> - [3] B Joy, et al. "Gender Shades: Intersectional Accuracy Disparities in Commercial Gender Classification", PMLR 2018.
>
> ### Typos
>
> **Response:** We have corrected the typo.

---

### Review · Reviewer_LiCi · 2026-03-16

**Summary Of Contributions:**

**Summary:**
The article addresses the intersection of robustness and fairness in image classification. More concretely, there is a claimed research gap regarding methods that not only produce well-performing and fair classifiers on curated data, but also generalize to naturally corrupted data. The contributions addressing such a gap are threefold:
- A new metric, called corrupted degradation disparity, quantifies performance decreases across two demographic groups by comparing the performance within each group on clean and corrupted test data. A greater disparity indicates greater perceived unfairness because the performance decrease in one group is much larger than that in the other group.
- FairSAM, an extension of the vanilla sharpness-aware maximization (SAM), reformulates the SAM-based optimization objective to balance gradient updates across groups. Intuitively, a corresponding reweighting mechanism introduces instance-wise weights such that each group has the same weight in the sum of instance-wise losses.
- An empirical evaluation study across the three face recognition datasets CelebA, FairFace, and LFW demonstrates improved fairness (lower corruption-induced degradation disparity), while maintaining competitive generalization performance (test accuracy).

**Strengths:**
- **[S1]:** The reweighting mechanism of FairSAM is a lightweight and intuitive approach that extends the well-established vanilla SAM objective to a group-aware fairness setting. Up to the constant $c$ as part of the initialization of the sample weights, no additional hyperparameter to the vanilla SAM is introduced.
- **[S2]:** The definition of the metric of corrupted degradation disparity is reasonable and intuitive, facilitating a broader adoption in upcoming evaluations.
- **[S3]:** The empirical evaluation study demonstrates clear reductions regarding the corrupted degradation across different settings, i.e., datasets with varying target/sensitive attributes and noise corruption levels. Furthermore, these results also justify FairSAM's design by comparing it to several other SAM variants.
- **[S4]:** A limitations paragraph clearly communicates the major drawback of addressing only image classification with only two groups defined through binary sensitive attributes.

**Weakness:**
- **[W1]:** While the limitations paragraph notes the scope restriction to image classification, the article's title implicitly makes a broader claim by referring to "Fair Classification". The article's scope is even narrower because all three image datasets are from the domain of facial recognition.
- **[W2]:** The results refer to a default hyperparameter setup. However, in practice, one would optimize hyperparameters by forming a corresponding validation split from the clean training data. Moreover, this default setup does not reveal how the choice of the hyperparameters specific to FairSAM affects its performance.
- **[W3]:** The evaluation is limited to ResNet-18 as the only neural network architecture. In practice, a common choice would be a more recent, pre-trained foundation model for image classification, e.g., DINOv3 [1].
- **[W4]:** While the image corruption follows the established ImageNet-C protocol, there is no consideration of different noise levels across groups. For example, if instances of one group have a higher noise level than instances of another group.
- **[W5]:** Beyond the clear motivation for the comparison with other SAM variants, it is unclear how the selection of the other competitors was made and why other fairness algorithms were not evaluated, e.g., CUMA [2] or DRO [3].
- **[W6]:** While the article indicates that a code appendix will be available (likely upon acceptance), no such code is available for review.

**Questions:**
- **[Q1]:** Is $\\rho=0.05$ instead of $\\epsilon=0.05$ meant in the sentence above Table 1?
- **[Q2]:** Should the constraint in Eq. (8) be $\\gamma_i > 0$ instead of  $p_i > 0$?
- **[Q3]:** To what does the variable $p$ refer in $l_{i,p}$ after Eq. (11)?

**References:**
- [1] Siméoni, Oriane, Huy V. Vo, Maximilian Seitzer, Federico Baldassarre, Maxime Oquab, Cijo Jose, Vasil Khalidov et al. "DINOv3." arXiv preprint arXiv:2508.10104 (2025).
- [2] Sagawa, Shiori, Pang Wei Koh, Tatsunori B. Hashimoto, and Percy Liang. "Distributionally robust neural networks for group shifts: On the importance of regularization for worst-case generalization." In International Conference on Learning Representations (2020).
- [3] Wang, Haotao, Junyuan Hong, Jiayu Zhou, and Zhangyang Wang. "How Robust is Your Fairness? Evaluating and Sustaining Fairness under Unseen Distribution Shifts." Transactions on Machine Learning Research (2023).

**Additional Comments:**

- **[C1]:** $\\gamma$ seems to refer in fact to a vector $\\boldsymbol{\\gamma}$ with a weight for each sample in Eq. (8). Similarly, it seems that the bold notation would also be more appropriate for the vector $\\boldsymbol{\\epsilon}$. Similar cases of inconsistent bold notation follow directly after Eq. (11) and in Eq. (12).
- **[C2]:** The typesetting of the quotation marks is partially off, e.g., in the third paragraph of the introduction.

**Audience:**

Yes

**Audience Explanation:**

Well-performing classifiers, which are not only fair for clean data but also for corrupted data, which often arise in a deployment phase, are of **high interest** for the research community and practitioners.

**Claims And Evidence:**

No

**Claims Explanation:**

While acknowledging the above listed strengths of the article, some weaknesses reveal that certain claims need more evidence. For example, the weakness **[W1]** indicates that the title does not accurately reflect the results, which show only superior evidence of FairSAM for face recognition datasets. Moreover, the data corruption is mainly synthetic. As mentioned in weakness **[W4]**, more complex noise schemes may be necessary to strengthen the claim of robustness under data corruption.

**Requested Changes:**

**Resolving the weaknesses** is the most important part. This can either be done by

- adjusting the claims, e.g., the title to resolve [W1],
- adding further experiments, e.g., experiments for other noise setups to resolve [W4],
- or explaining that a weakness is in fact a misunderstanding from my side.

Furthermore, providing sufficient answers to the questions and resolving the additional (minor) comments at the end of this review would further strengthen the article. I'm looking forward to a productive discussion phase.

---

> ### Author Response · Authors · 2026-04-24
> **Response to Reviewer LiCi**
>
> We sincerely thank the reviewer for the thorough evaluation and detailed feedback that significantly improved our manuscript. Below is our response to each comment.
>
> ### W1: Scope restriction to facial recognition datasets
>
> **Response:** This is an excellent point. We have expanded our evaluation to include **CheXpert**, a large-scale medical imaging dataset for chest X-ray diagnosis. CheXpert experiments (Table 4) evaluate fairness in pleural effusion detection across gender subgroups. Results demonstrate that FairSAM achieves the lowest corrupted degradation disparity and performance disparity while maintaining the highest accuracy. This validates that our framework generalizes effectively beyond facial recognition to medical imaging domains where fairness and robustness are also critical.
>
> ### W2: Default hyperparameter setup
>
> **Reviewer:** "The results refer to a default hyperparameter setup. However, in practice, one would optimize hyperparameters by forming a corresponding validation split from the clean training data."
>
> **Response:** We appreciate this clarification. The hyperparameters we use (ρ = 0.05, c = 1) are established defaults from the original SAM paper, which identified these values as optimal through extensive validation across multiple datasets. These values are widely adopted in the SAM literature and have proven robust across diverse settings.
>
> ### W3: Evaluation limited to ResNet-18
>
> **Response:** We have added comprehensive experiments with **DINOv3**, a state-of-the-art vision foundation model. Table 5 presents results on FairFace using DINOv3. Results show that FairSAM with DINOv3 achieves superior fairness metrics compared to other baselines.
>
> ### W4: Consideration of different noise levels across groups
>
> **Response:** We have added experiments with **asymmetric noise corruption** where the advantaged group (s⁺) experiences level-4 Gaussian noise while the disadvantaged group (s⁻) experiences level-5 Gaussian noise (Table 5). FairSAM demonstrates robust performance under asymmetric noise. This confirms that our fairness-aware mechanism effectively balances robustness across groups even when corruption severity differs.
>
> ### W5: Selection of competitors
>
> **Response:** We have added **GroupDRO (Distributionally Robust Optimization)** and **Momentum-SAM (MSAM)** as additional competitive baselines. GroupDRO is specifically designed to optimize worst-group performance and MSAM is a recent SAM variant designed for efficiency. New results are now included in all main experimental tables (Tables 1-5)
>
> ### W6: Code availability
>
> **Response:** We have added a code repository link to ensure reproducibility.
>
> ### Q1: Is ρ=0.05 instead of ϵ=0.05 meant?
>
> **Response:** We have fixed this notation error.
>
> ### Q2: Should the constraint in Eq. (8) be γᵢ>0 instead of pᵢ>0?
>
> **Response:** We have fixed this typo.
>
> ### Q3: To what does the variable p refer in lᵢ,ₚ after Eq. (11)?
>
> **Response:** We have corrected this notation with $\gamma$ instead of $p$.

---

> > ### Comment · Reviewer_LiCi · 2026-05-10
> > **Response to Manuscript Update:**
> >
> > Dear Authors,
> >
> > First of all, many thanks for your reply and the corresponding updates to the manuscript. Moreover, sorry for my delayed response. Overall, your updates notably improve your manuscript. In the following, I detail my impression of these updates regarding the weaknesses of my original review:
> >
> > - **Weaknesses [W3], [W4], [W5], and [W6] are fully resolved**. As a result, the robustness of FairSAM is confirmed for a state-of-the-art backbone [W3], varying noise levels within subgroups [W4], and in comparison to other fairness algorithms [W5]. Moreover, the available codebase [W6] makes it easier to use FairSAM in future applications or research.
> > - **Weaknesses [W1] and [W2] are partially resolved**. In particular, the addition of CheXpert as another image dataset is a significant improvement on the claims [W1], as it demonstrates superior performance in the medical domain. Yet, the restriction to image data remains. Since this restriction is noted in the limitations, it is largely acceptable, though a title update could better align with your empirical claim. However, I also understand if you want to keep your title since FairSAM can, in principle, be applied to any other data modality. The rationale for using established default hyperparameters [W6] is reasonable, as it demonstrates performance under limited computational resources. Still, results for more computational resources enabling hyperparameter optimization to see (1) the effect of the hyperparameters and (2) the potential tuning capabilities of the FairSAM approach would have been another good addition.
> >
> > A new aspect for discussion is the **extension to multi-class and multi-sensitive attributes**. In general, I agree with my colleague reviewers that such an extension would be a significant improvement to the manuscript. However, there are a few issues with the current form of this update:
> > - **[I1]:** Section 3.1 introduces the general problem of fair classification for binary sensitive attributes. In alignment with this, Definition 1 introduces the corrupted degradation disparity for binary-sensitive attributes as well. The subsequent extension to multi-class and multi-sensitive attributes feels off from a "reading flow perspective" because it lacks a formal definition (e.g., Definition 2) and does not align with your problem setting. A more consistent writing would be to introduce the general setting and then say the binary setting is just a special case.
> > - **[I2]:** A brief explanation of why the worst-case disparity is better than alternatives (e.g., average) is missing.
> > - **[I3]:** If I understood the meaning of $\\mathcal{S}$ correctly, it refers to all potential combinations of attributes and their values as distinct groups. I wonder whether, with an increasing number of attributes and values per attribute, we face combinatorial complexity problems, since there can be many such groups. As a result, the number of samples per group can be quite low, which can not only affect FairSAM's performance but also the stability of the metric.
> > - **[I4]:** As noted in the limitations, there are no results for this extended setting.
> >
> > Overall, these issues should be addressed. As an alternative to improving textual embedding [I1, I2] with additional experiments [I4], I would move this extension to the limitations section, where you have already pointed out issue [I4]. Then, this extension feels less like one of your main claims (which you currently do not validate), but you already show a possible extension without making it a part of your methodological claims.
> >
> > Finally, there are a few remaining and new (including ones I missed in my original review) **editorial remarks**:
> > - You could explain that in Eq. (3) $|\\cdot|^{q-1}$ denotes the absolute power and value per vector element.
> > - Some quotation marks are still a bit off, e.g., ``flatter¨ in Section 3.2.
> > - A comma is missing after Eq. (10).
> > - A dot is missing after Eq. (11).
> > - Instead of an unpaired footnote for the source code, you could write "All source code is available at our repository^1".
> > - Is there a reason to sort the approaches across tables in different orders?
> > - In Section 5.4, do you actually evaluate model robustness under asymmetric noise corruption across two backbone architectures, or do you mean for an additional backbone? I have this question because in this subsection, you only refer to Table 5.4, which contains results only for the DINOv3 backbone. In this context, naming the exact DINOv3 model variant would also be a good addition.

---

> > > ### Author Response · Authors · 2026-05-14
> > >
> > > Dear Reviewer LiCi,
> > >
> > > Thank you for your constructive follow-up.
> > >
> > > We agree with your comments and have revised the manuscript to better align its scope with the empirical evidence. We changed the title to "FairSAM: Fair Classification on Corrupted Image Data Through Sharpness-Aware Minimization." We also clarified that the current study focuses on image classification with binary sensitive attributes.
> > >
> > > We agree that multi-class classification and multiple sensitive attributes are important extensions. Our metric can naturally handle multi-class tasks because it is based on accuracy and accuracy disparity. We note that most fair classification studies focus on binary sensitive attributes, while extensions to multiple sensitive attributes constitute a well-established and relatively independent research direction. Since this paper focuses on the SAM formulation and optimization, we moved this extension from the main methodology section to the limitations and future work section.
> > >
> > > We also addressed the editorial comments. Specifically, we clarified the notation after Eq. (3), corrected the quotation marks, added the missing punctuation after Eqs. (10) and (11), revised the source-code footnote, and made the ordering of methods consistent across tables. We also clarified in the experimental settings that the DINOv3 backbone uses the ViT-S/16 distilled variant.
> > >
> > > Finally, we revised Section 5.4 to clarify the backbone comparison. We now report results for both DINOv3 and ResNet18 under asymmetric noise corruption.
> > >
> > > We are currently running one additional experiment, in which we conduct a grid search over values of $c$ and $\rho$ to study FairSAM’s hyperparameter sensitivity and tuning potential. Due to the time limit, the results are not yet available. However, we will include them in the next version.

---

> > > > ### Comment · Reviewer_LiCi · 2026-05-19
> > > >
> > > > Dear Authors,
> > > >
> > > > Again thanks for your fast revisions, such that your manuscript better reflects your actual contributions. Your plan to add a small hyperparameter study is also welcome and potentially helps other researchers or practitioners with the adoption of FairSAM.

---

### Review · Reviewer_EWsw · 2026-04-09

**Summary Of Contributions:**

This paper focuses on fairness under data corruption, highlighting that robustness methods such as SAM can lead to uneven performance degradation across demographic subgroups. To address this, the authors propose a new metric, Corrupted Degradation Disparity, and introduce FairSAM, which integrates fairness into SAM via instance reweighting. Experiments show that this method has certain effectiveness. The authors are recommended to address the following comments to improve the significance of the study and for the benefit of a wider audience.

1. The metric $\Delta p$ is defined only for binary attributes and lacks a formal extension to multi-class settings.

2. With the rapid advancement of LLMs [1] and generative models [2-5], fairness issues have also emerged in these domains. However, the Related Work does not cover these recent developments. It is recommended that the authors include a discussion on the potential applicability of this work to large-scale models.

3. The baselines used in this paper are relatively outdated. It is suggested to include comparisons with 1-2 more recent SAM-related methods.

[1] A survey on fairness in large language models.

[2] Fair diffusion: Instructing text-to-image generation models on fairness.

[3] Finetuning text-to-image diffusion models for fairness.

[4] Balancing act: Distribution-guided debiasing in diffusion models.

[5] LightFair: Towards an efficient alternative for fair T2I diffusion via debiasing pre-trained text encoders.

**Audience:**

Yes

**Audience Explanation:**

The paper addresses the intersection of robustness and fairness under data corruption, which is a timely and important topic for the TMLR community. Its findings are relevant to researchers working on reliable and trustworthy machine learning in real-world deployment settings.

**Broader Impact Concerns:**

None.

**Claims And Evidence:**

Yes

**Claims Explanation:**

The main claims of the paper are supported by clear, accurate, and convincing evidence. The experiments clearly demonstrate improvements in both robustness and fairness metrics compared to relevant baselines.

**Requested Changes:**

Please address the issues raised in the Summary of Contributions section.

---

> ### Author Response · Authors · 2026-04-24
> **Response to Reviewer EWsw**
>
> We thank the reviewer for the positive evaluation and constructive suggestions. Below is our response to each comment.
>
> ### Comment 1: Multi-class metric extension
>
> **Response:** We have added a formal extension to multi-class classification and multi-sensitive attributes in Section 4.1 (highlighted in yellow). Our fairness metric operates on any performance measure $M$ and extends naturally to multi-class problems by substituting appropriate metrics (accuracy, F1-score, etc.) into the degradation calculation.
> Regarding multi-sensitive attributes, we extend the formulation to quantify the largest performance gap between any two demographic groups under corruption. The extension is provided in Equations (1)-(2) in the revised manuscript.
>
> ### Comment 2: Discussion on applicability to large-scale models
>
> **Response:** We acknowledge this is an important direction. While our current work focuses on image classification with discriminative models, the principles of fairness-aware robustness optimization extend conceptually to generative models and LLMs. We have added a brief discussion in the Related Work section acknowledging recent fairness concerns in generative AI.
>
> ### Comment 3: More recent SAM-related baselines
>
> **Response:** We have added two recent SAM-related baselines: **MSAM (Momentum-SAM)** and **GroupDRO (Group Distributionally Robust Optimization)**. These methods are now included in all experimental tables (Tables 1-5). MSAM represents recent advances in SAM variants, while GroupDRO is specifically designed to address uneven robustness concerns. Our results show that FairSAM consistently outperforms both methods in terms of corrupted degradation disparity while maintaining competitive accuracy.

---

### Decision · Action_Editor_7PCo · 2026-05-25

**Recommendation:** Accept with minor revision

**Additional Comments:**

Please include the following revisions in the final version:

- Include the promised hyperparameter sensitivity study (or the corresponding results if already completed).
- Ensure that the final manuscript fully complies with the TMLR formatting and submission requirements.

**Audience:**

Yes

**Audience Explanation:**

The paper addresses fairness and robustness under corrupted image data, which is relevant to researchers working on trustworthy machine learning, robust optimization, and fairness in real-world deployment settings. This topic is of clear interest to the TMLR audience.

**Claims And Evidence:**

Yes

**Claims Explanation:**

The submission investigates fairness under distribution corruption in image classification and proposes FairSAM, an extension of Sharpness-Aware Minimization (SAM) that incorporates fairness-aware reweighing to reduce performance disparity across demographic subgroups under corrupted inputs. The paper also introduces a metric to quantify subgroup degradation under corruption and evaluates the method on multiple vision datasets and backbones.

Reviewers were generally positive, agreeing that the topic is timely and relevant to TMLR. FairSAM was viewed as a meaningful SAM extension with empirical gains in robustness and fairness.

In response to the reviews, the authors have made several improvements:

- The authors addressed most concerns by expanding the evaluation with CheXpert, DINOv3, asymmetric-noise experiments, stronger baselines, and a released code repository.
- They strengthened the theoretical framework by extending the fairness metric to multi-class and multi-sensitive-attribute settings and adding formal definitions for intersectional demographic groups.
- They enhanced the experimental evaluation by adding GroupDRO and MSAM as competitive baselines, with clearer descriptions and proper citations for all baselines.
- Sections 4.2 and 4.3 were substantially revised for clarity and logical flow, and notation inconsistencies were fixed.
- The authors refined the scope by revising the title, clarifying the focus on corrupted image classification with binary sensitive attributes, and moving the multi-sensitive-attribute extension to the limitations/future work section to better align claims with evidence.

Overall, most major concerns have been addressed satisfactorily through revisions and clarifications. One minor remaining issue is the pending **hyperparameter study**. The authors should add the promised grid search over $\rho$ and $c$ to study FairSAM’s hyperparameter sensitivity and tuning potential in the **final revision**. This addition was requested by the reviewer and contributed to the updated positive recommendation.